# Rationally Designed CdS-Based Ternary Heterojunctions: A Case of 1T-MoS_2_ in CdS/TiO_2_ Photocatalyst

**DOI:** 10.3390/nano11010038

**Published:** 2020-12-25

**Authors:** Wenqian Chen, Shaomei Zhang, Ganyu Wang, Gang Huang, Zhichong Yu, Yirui Li, Liang Tang

**Affiliations:** 1Key Laboratory of Organic Compound Pollution Control Engineering, Ministry of Education, Shanghai 200444, China; shaomeizhang@shu.edu.cn (S.Z.); wgy@shu.edu.cn (G.W.); zhichongyu@shu.edu.cn (Z.Y.); yiruili@shu.edu.cn (Y.L.); 2Shanghai Institute of Applied Radiation, Shanghai University, 20 Chengzhong Road, Shanghai 201800, China; 3School of Environmental and Chemical Engineering, Shanghai University, Shanghai 200444, China; 4Physical Science and Engineering Division King Abdullah University of Science and Technology (KAUST), Thuwal 23955-6900, Saudi Arabia; gang.huang@kaust.edu.sa

**Keywords:** heterojunction, photocatalyst, hydrogen evolution, CdS/1T-MoS_2_/TiO_2_

## Abstract

As promising heterojunction photocatalysts, the binary CdS-based heterojunctions were investigated extensively. In most of the reported CdS-based heterojunctions, however, electrons come from the semiconductor with wide band gap (e.g., TiO_2_) would limit the visible-light absorption of CdS and hence lower the performance. In this work, we introduced 1T-MoS_2_ to form a novel ternary heterojunction, namely CdS/1T-MoS_2_/TiO_2_, in which 1T-MoS_2_ has more positive conduction band than CdS and TiO_2_. The hydrogen evolution rate of CdS/1T-MoS_2_/TiO_2_ reaches 3.15 mmol g^−1^ h^−1^, which is approximately 12 and 35 times higher than that of pure CdS and CdS/TiO_2_ binary heterojunction under the same conditions, respectively. This performance enhancement could be attributed to the presence of 1T-MoS_2_ and a plausible mechanism is proposed based on photoelectrochemical characterizations. Our results illustrate that the performance of CdS-based heterojunctions for solar hydrogen evolution can be greatly improved by appropriate materials selection.

## 1. Introduction

Semiconductor-based photocatalyst have been widely employed in the water splitting and treatment of various environmental pollutants [1]. Photocatalytic water splitting using semiconductor heterojunctions has gathered great interest recently, mostly due to efficient utilization of visible light [2]. In the past few decades, CdS-based heterojunctions have investigated extensively due to the narrow band gap of CdS (2.4 eV) [3]. Furthermore, the conduction band potential of CdS is more negative than the reduction potential of protons. Hence, CdS has a sufficient potential to reduce protons into H_2_ [4,5]. The CdS-based photocatalysts have been extensively explored for the hydrogen evolution due to its suitable band gap structure and strong capacity of light absorption. However, CdS suffer from the problem of photo-corrosion wherein S^2−^ ions get easily oxidized by photogenerated holes [6]. The strategy to overcome the photo-corrosion of CdS is therefore highly desirable. Solutions including anion substitution [7,8] and the formation of a heterojunction have been proposed. In contrast to the anion substitution, combing a semiconductor with wide band gap to form heterojunctions is an alternative which is synthetically easier.

Numerous wide band gap semiconductors have been used in combination with CdS to form a Z-type heterojunction to overcome the photocorrosion of CdS by photo-generated holes h^+^ [9]. Among them, TiO_2_ with a band gap of ca. 3.2 eV is good choice [10]. On the one hand, both materials are cheap for practical applications. On the other hand, the CdS with narrower band gap could use visible light more efficient. However, deeper valence band (VB) of CdS could allow the transfer of electrons from TiO_2_-CB (conduction band) to CdS-VB, as illustrates in Scheme 1a. Hence, under a visible light irradiation, the binary heterojunction CdS/TiO_2_ cannot effectively solve the problem of easy recombination of electrons and holes, which reduces the catalytic activity. Therefore, a third component with more positive CB to form a ternary heterojunction is a possible solution [11], as shown in Scheme 1b.

For the third component, many compounds have been tested at several concentration ranges, such as WO_3_ [11], ZnIn_2_S_4_ [12], SiO_2_ [13], and Fe_3_O_4_ [14]. Herein, we focus on the MoS_2_, which belongs to the family of layered transition metal dihalides [15,16,17,18]. MoS_2_ has four morphologies: Twisted tetragonal phase (1T), hexagonal phases (1H and 2H), and rhombic phase (3R) [19]. The semiconductor phase 2H-MoS_2_ is the most stable phase. 1T-MoS_2_ (metal phase) can usually be obtained by chemical exfoliation of 2H-MoS_2_ and embedding of various ions (Li^+^, Na^+^, K^+^) [20]. The conductivity of 1T phase is 10^7^ times higher than 2H phase [21], which facilitates the rapid flow of charge and improves the reactivity and hence were investigated widely for photocatalytic water splitting.

In this work, the CdS/1T-MoS_2_/TiO_2_ ternary heterojunction was synthesized by hydrothermal method. We discovered that the introduction of two-dimensional sheet-like 1T-MoS_2_ can accelerate the charge transfer and increase the rate of photocatalytic hydrogen production, and the use of nanosheet 1T-MoS_2_ coated around the CdS nanoparticles can prevent them from photocorrosion. Through a specific hybridization method and the energy band modification of each substance, an internal electric field is constructed to form the directional transfer of photo-generated carriers. The photocatalytic hydrogen activity reached 3.15 mmol g^−1^ h^−1^ (35 times that of binary CdS/TiO_2_), indicating an important role of 1T-MoS_2_ in the CdS-based heterojunctions.

## 2. Materials and Methods

### 2.1. Preparation of Photocatalysts

The CdS/TiO_2_ binary heterojunction was synthesized by a hydrothermal method with anatase TiO_2_ nanoparticles. TiO_2_ was prepared by solvothermal method, as previously reported [22]. The Cd(CH_3_COO)_2_·2H_2_O (AG, Adamas, 1.38 mmol) and thiourea (AG, Adamas, 5.52 mmol) were dispersed in 35 mL pure water, then the obtained TiO_2_ was added, the molar ratio of TiO_2_ is 15%, 25% and 35% of the Cd amount. Stir evenly and transfer to the 50 mL Teflon-lined stainless-steel autoclave. The mixture was reacted at 453 K for 5 h, followed cooled to room temperature naturally. The samples were washed with pure water and ethanol for several times, then vacuum dried at 333 K overnight. Bare CdS was synthesized by the same method without TiO_2_ [23].

The ternary CdS/1T-MoS_2_/TiO_2_ heterojunction was synthesized with the addition of CTAB and (NH_4_)_6_Mo_7_O_24_·4H_2_O. First, CdS (0.1444 g) was added into 50 mL pure water. 10 mL of CTAB (AR, Sinopharm Chemical Reagent Co., Ltd. Shanghai, China) aqueous solution (0.01 M) was then added into above orange suspension. (NH_4_)_6_Mo_7_O_24_·4H_2_O (AG, Adamas) and thiourea (n(Mo):n(S) = 7:15, AG, Adamas) were further added to the above mixture with the Mo atoms are 25%, 35%, and 45% molar ratio of CdS. The mixture was transferred into 100 mL Teflon-lined stainless-steel autoclave, and reacted at 473 K for 24 h. The samples were centrifuged and washed with pure water and ethanol for several times, then vacuum dried at 333 K overnight. Second, the obtained binary heterojunction “CdS/1T-MoS_2_-45” (45% of MoS_2_) was put into 45 mL ethanol (AR, Adamas) and stirred to form suspension A, the corresponding amount of TiO_2_ (the molar amount is 15%, 25%, and 35% of CdS) was then added into 15 mL ethanol and ultrasonic for 2 h to form suspension B, then suspension B was poured into suspension A. The mixture was transferred to 100 mL Teflon-lined stainless-steel autoclave, and reacted at 393 K for 2 h. The samples were obtained after washed and vacuum dried at 333 K overnight. The ternary products were named as CdS/1T-MoS_2_/TiO_2_-45-15, CdS/1T-MoS_2_/TiO_2_-45-25 and CdS/1T-MoS_2_/TiO_2_-45-35. The pure 1T-MoS_2_ can be obtained by (NH_4_)_6_Mo_7_O_24_·4H_2_O and thiourea in accordance with the above proportion {n(Mo):n(S) = 7:15} and reaction conditions [24].

### 2.2. Characterization

X-ray diffraction (XRD) patterns were obtained from DX-2700 X-ray diffractometer with Cu-Kα radiation operated at 40 kV and 40 mA. Scanning electronic microscopy (SEM) images were conducted on ZEISS MERLIN Compact. Transmission electron microscopy (TEM) and high-resolution transmission electron microscopy (HRTEM) observations were performed on a JEOL JEM-2010F electron microscope operating at 200 kV. X-ray photoelectron spectroscopy (XPS) was tested on Thermo ESCALAB 250XI using Al-Kα excitation source (*hν* = 1486.6 eV) and C_1s_ = 284.60 eV combined energy standard was used for charge correction. Raman spectra were detected by a Japan HORIBA system with a 532 nm laser. UV-vis diffuse reflectance spectra (DRS) were carried out on a UV-vis spectrometer (UV-2600, Shimadzu, Japan), BaSO_4_ as whiteboard to deduct background value. The photoluminescence (PL) spectra were measured by a F-320 Fluorescence Spectrophotometer that excitation wavelength is 494 nm.

### 2.3. Photocatalysis

The photocatalytic reaction was conducted on an all glass automatic on-line trace gas analysis system (Labsolar-6A) and SHIMAZU gas chromatograph (GC-2014C). The hydrogen generated in the reactor was automatically entered into the gas chromatograph for detection. Fifty milligrams of catalyst was placed in the reactor and a 100 mL aqueous solution containing 10 mL of lactic acid was poured into the reactor. The reactor temperature was kept at 278 K by a low temperature thermostatic bath. The light source is a 300 W Xe lamp (λ ≥ 350 nm).

### 2.4. Photoelectrochemical Measurements

Mott–Schottky analysis, ampere i-t curve analysis, and electrochemical impedance spectroscopy (EIS) were performed by electrochemical workstation (CHI660E instrument). The electrodes were prepared by spin-on-conductive glass process, as previously reported [25]. A three-electrode system with the electrolyte of 0.5 M Na_2_SO_4_ solution was used, in which the counter electrode was platinum, the reference electrode was saturated calomel electrode (SCE), and the working electrode was conductive glass coated with photocatalyst. The Mott–Schottky measurement was measured using an impedance-potential model with a voltage range of −1.5~1.5 V. The amperometric i-t curve was measured by a 300 W Xe lamp (CEL-HXF300). The EIS test was performed at a frequency range between 10^−2^ and 10^5^ Hz.

## 3. Results and Discussion

### 3.1. Structural Analysis

Figure 1 presents the XRD patterns of the binary and ternary CdS heterojunctions. XRD pattern of the bare CdS sample is consistent with the hexagonal standard CdS card (JCPDS card no. 89-2944) (Appendix A). SEM image shown the CdS is formed by the aggregation of small nanoparticles with the particle size of ca. 100 nm (Appendix A). Appendix A shows the XRD and SEM topography of TiO_2_. It can be seen that the TiO_2_ is a square nanosheet of about 50 nm. Appendix A provides the structural analysis of 1T-MoS_2_ nanosheets. The 1T-MoS_2_ nanosheets in this work were grown in situ on the surface of CdS nanoparticles, forming a binary heterojunction CdS/1T-MoS_2_. Appendix A shown the characteristic peak of binary heterojunction is the CdS, and there is no 1T-MoS_2_ peaks were observed. This could be due to the low crystallinity of 1T-MoS_2_. Moreover, the formation of CdS/1T-MoS_2_ do not change the structure of CdS. A similar XRD pattern of binary CdS/TiO_2_ (Appendix A) and ternary heterojunctions were observed. Because the 2theta value of the TiO_2_ (101) crystal plane and the CdS (100) crystal plane are very small, the corresponding peak of TiO_2_ is not visible in Appendix A, but it is obvious that the peak intensity around 25° is significantly enhanced. This indicates that TiO_2_ and 1T-MoS_2_ only interacts with the surface of CdS and does not change the crystal structure of each material. These results have confirmed the successful loading of the molybdenum disulfide and titanium dioxide on the CdS surface.

As shown in Figure 2a, the ternary heterojunctions are clustered together in the form of nanoparticles. At the edge of the particles, unlike pure CdS (Appendix A), it can be clearly seen that the transparent flakes 1T-MoS_2_ wrap the CdS nanoparticles, and the small square nanosheets TiO_2_ are scattered and uniformly distributed on the flakes. By magnifying the edges of particles, a high-resolution transmission electron microscopy (HRTEM) is obtained. The corresponding diffraction pattern can be obtained by fast Fourier transform (FFT) at the lattice fringe of HRTEM. The distance from the diffraction spot to the center is consistent with the lattice fringe spacing on the HRTEM image. The lattice spacing calculated from FFT also matches the corresponding d value of the XRD pattern. The lattice spacing marked 2.48 Å, 9.32 Å, and 3.48 Å in the HRTEM image (Figure 2b), which respectively represent (102) crystal planes of CdS, (002) crystal planes of 1T-MoS_2_, and (101) crystal planes of TiO_2_. The close contact between the three substances is beneficial to the rapid transfer of photogenerated electrons, and thus facilitating the production of hydrogen. Moreover, compared to the 2H phase, the expanded layer spacing of 1T-MoS_2_ speeds up the electron flow this improves the catalytic performance of the heterojunctions. EDS element mapping (Figure 2c) shown the existence of Cd, S, Mo, Ti, O and their uniform distribution in the composites. In addition, EDS spectrum (Appendix A) provides the semi-quantitative value of each element in the sample of CdS/1T-MoS_2_/TiO_2_-45-15. This indicates that the main content in the heterojunction is Cd and S, while other elements exist in a small amount. This EDS result agreement well with the XRD patterns that the characteristic peaks of both binary and ternary heterojunctions are consistent with that of CdS.

Raman spectroscopy and X-ray photoelectron spectroscopy (XPS) were conducted for further analysis of the ternary heterojunctions, as shown in Figure 3. The characteristic peaks at 299 cm^−1^ and 597 cm^−1^ in CdS/1T-MoS_2_/TiO_2_-45-15 are attributed to the longitudinal optical phonon mode of CdS in the Raman spectra [26]. When CdS was coupled with 1T-MoS_2_ and TiO_2_, 150 cm^−1^ corresponding peaks of 1T-MoS_2_ and 397 cm^−1^ corresponding peaks of TiO_2_ are observed in the Raman spectrum of the composite CdS/1T-MoS_2_/TiO_2_-45-15 (Figure 3a), indicates the CdS has a close interaction with other two compounds. Appendix A shown the XPS full spectrum of ternary heterojunction, which involves the composite elements of Cd, S, Mo, Ti, and O. Figure 3b shown the peaks at 404.69 eV and 411.43 eV of CdS/1T-MoS_2_/TiO_2_-45-15 heterojunction represent the 3d_5/2_ and 3d_3/2_ orbits of Cd^2+^, respectively. Compared to the bare CdS (404.81 eV and 411.55 eV) [27], the low energy peaks in the heterojunctions may be due to the interaction of CdS with 1T-MoS_2_ and TiO_2_. Figure 3c shown the Mo peaks of 229.82 eV and 232.92 eV in the heterojunction are respectively represented by the 3d_5/2_ and 3d_3/2_ of Mo^4+^ [28]. Compared to the bare 1T-MoS_2_, the peaks are shifted to the high energy, indicating the change of electronic density around Mo element. After deconvoluting the Mo 3d environment (Appendix A), the 1T phase content remains high, and an additional Mo-O bond peak appears at 233.19 eV and 235.46 eV [29,30]. As shown in Figure 3d, when the binary heterojunction CdS/1T-MoS_2_-45 was coupled with TiO_2_, the 2p orbital energy of Ti element is also shifted, indicates the interaction between TiO_2_ and 1T-MoS_2_. Therefore, Raman and XPS results further revealed the successful synthesis of CdS/1T-MoS_2_/TiO_2_ hetero-structured composites.

### 3.2. Photocatalysis

The catalysts were photocatalyzed to produce hydrogen under a simulated sunlight irradiation (λ ≥ 350 nm), and lactic acid was used as a reversible electron donor. Figure 4 and Appendix A shown the hydrogen production of different materials. It can be seen from Appendix A that the hydrogen production increases linearly with the irradiation time increased. Figure 4a shown the activity was greatly enhanced with the combined of 1T-MoS_2_ in the binary CdS/TiO_2_ heterojunctions. Interestingly, the activity of binary CdS/TiO_2_ heterojunctions is lower than that of bare CdS, suggesting an electron transfer limitation between the wide band gap TiO_2_ and CdS. Moreover, the best composition of ternary heterojunctions is CdS/1T-MoS_2_/TiO_2_-45-15. We will discuss the relationship between the activity and composite differences in the later section. Figure 4b provides the three times of cycle hydrogen evolution experiments on CdS/1T-MoS_2_/TiO_2_-45-15, indicating a stable ternary heterojunction semiconductor-based photocatalyst was obtained. Furthermore, the structure of CdS/1T-MoS_2_/TiO_2_-45-15 was maintained after photocatalytic reactions (Appendix A) as evidenced by XRD.

### 3.3. Light Absorption Properties

UV-visible DRS spectra of the heterojunctions are shown in Figure 5a. The heterojunction has absorption edges at around 570 nm, which is close to the bare CdS and binary CdS/1T-MoS_2_. However, different with the CdS, the visible light absorption of binary and ternary heterojunctions was enhanced in the ranges of 570–780 nm, indicates the formation of heterojunction can promote the light absorption. Appendix A shown the effect of 1T-MoS_2_ and TiO_2_ concentrations on the light absorption of binary heterojunctions. Apparently, the absorbance increases after the addition of 1T-MoS_2_, which could partly explain the hydrogen production gradually increases as its loading amount increased in heterojunctions (Figure 4a). However, the absorbance did not increase significantly when TiO_2_ amounts were increased. It is due to the weak light absorption ability of white TiO_2_. This suggest the addition of 1T-MoS_2_ plays a key role in the enhanced light absorption properties.

### 3.4. Charge Carrier Dynamics

Photoluminescence (PL) spectra and photocurrent response were performed on the catalysts to further study the dynamics of photogenerated electron-hole pairs. As shown in Figure 5b, it is clear that CdS has a relatively high PL intensity. The appearance of 1T-MoS_2_ will weaken the PL intensity. This result is supported by the hydrogen release amount of different samples of binary heterojunction CdS/1T-MoS_2_. Afterwards, the addition of TiO_2_ greatly weakened the PL intensity, indicating that the photogenerated carriers were largely captured by TiO_2_, thereby inhibiting the recombination of electron-hole pairs. The EIS Nyquist plots of the catalysts were shown in Figure 5c. It is well known that the smaller the EIS spectra the lower of the charge-transfer resistance, and hence an enhanced electron transfer capability and high separation effectiveness of the photogenerated electron-hole pairs. Compared to the bare CdS and binary CdS/1T-MoS_2_, the ternary CdS/1T-MoS_2_/TiO_2_-45-15 contain a quite small arc radius, indicates the low impedance and hence high charge separation efficiency in the ternary heterojunctions. The photocurrent response (Figure 5d) can also further illustrate the efficiency of photogenerated charge carrier separation. After repeated on-off light irradiation for several times, the response current of CdS/1T-MoS_2_/TiO_2_-45-15 is strong and relatively stable, which is consistent with the experimental data of PL, EIS and hydrogen evolution experiments. These results clearly indicate the presence of 1T-MoS_2_ in the CdS/TiO_2_ binary heterojunction promotes the electron transfer and reduces the recombination of electron hole pairs.

### 3.5. Band Potential Analysis

We further analysis the band positions of three substances to explore the possible mechanism of ternary heterojunctions. Appendix A shown the UV-vis diffuse reflection spectrum of these three single substances. Use the diffuse reflection data to obtain (*αhν*)^1/2^ against *hν* plots and it provides the band gaps in Appendix A, where the band gaps of CdS and TiO_2_ are 2.02 eV and 2.93 eV, respectively. 1T-MoS_2_ is metallic with no band gap [31]. This further confirmed by XPS in Appendix A, which shown that the *E*_VB_ position of 1T-MoS_2_ is ~0 V.

Furthermore, the Mott–Schottky analysis was used to determine the flat band potential (*E*_FB_) of semiconductor materials. Appendix A shows the Mott–Schottky plots of CdS, TiO_2_, and 1T-MoS_2_ with the frequency ranges from 300–1000 Hz. In each case, the *C*^−2^ values decreased as the applied potential *E* became more negative. This is a typical behavior for an n-type semiconductor. The *E*_FB_ can be estimated by extrapolating the Mott–Schottky plot linearly to the *x* axis intercept. The *E*_CB_ positions of CdS, TiO_2_ and 1T-MoS_2_ were estimated as −1.20 ± 0.01 V, −0.97 ± 0.02 V, and −0.43 ± 0.02 V (vs. SCE), respectively (equivalent to −0.96 V, −0.73 V and −0.19 V vs. NHE). It is known that the conduction-band minimum (*E*_CBM_) is 0.1–0.3 V more negative than the *E*_FB_ in an n-type semiconductor [32]. Hence, the *E*_CBM_ values are located at ca. −1.06 V for CdS, −0.83 V for TiO_2_, and −0.29 V for 1T-MoS_2_ (vs. NHE). The resulting hydrogen production mechanism diagram is shown in Figure 6. It can be seen that the *E*_CBM_ of 1T-MoS_2_ is more positive than that of TiO_2_. Therefore, 1T-MoS_2_ is suitable for the third substance in CdS/TiO_2_ binary heterojunction. This is the origin of the high performances on ternary heterojunction.

## 4. Conclusions

This study set out an efficient CdS/1T-MoS_2_/TiO_2_ ternary heterojunction, synthesized by a simple hydrothermal method. The presence of 1T-MoS_2_ greatly promotes the photocatalytic property of binary CdS/TiO_2_ heterojunctions (35 times that of binary CdS/TiO_2_). This could be explained by the improvement of charge carrier dynamics (electron-hole pairs) and has been systematically discussed on the basis of PL, EIS, and photocurrent results. In addition, the metallicity and two-dimensional layered structure of 1T-MoS_2_ are conducive to accelerate the electron transfer, and its coating can avoid the photo-corrosion of CdS. This greatly enhance the catalytic performance of the material and thus has great potential for CdS-based heterojunctions.

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
