# Peer review of "Rationally Designed CdS-Based Ternary Heterojunctions: A Case of 1T-MoS2 in CdS/TiO2 Photocatalyst"

_nanomaterials, 2020, doi:10.3390/nano11010038_

Round 1

Reviewer 1 Report

W. Chen, L. Tang et al., reported the use of CdS/MoS2/TiO2 for the photocatalytic hydrogen evolution reaction. The author observed the interesting photocurrent of hydrogen evolution reaction. The concept seems interesting but potential for the hydrogen evolution reaction should be shown. RHE should be -0.42 V vs NHE, which is apparently more negative than conduction band edge of MoS2. I am just suspicious about the catalytic activity of MoS2 in this experiment. Author should note the reason for this experimental points.

I recommend this article should be minor revision.

Reviewer 2 Report

Wen-Qian Chen et al. reported ternary CdS/1T-MoS2/TiO2 heterojunction synthesized by a hydrothermal method for hydrogen evolution. The author has studied their characterization using XRD, TEM, UV-Vis spectroscopy, XPS analysis and EIS analysis. Although the characterizations are comprehensive, some of the experiment results need to be described in more detail. Overall, after appropriate revision, I could consider this manuscript to be published in Nanomaterials. The specific comments are shown below:

  1. Please provide FTT of HRTEM images for Cds, 1T-MoS2 and TiO2, and give more depiction in TEM image.
  2. For EDS analysis, why only selected CdS/1T-MoS2/TiO2-45-15 heterojunctions to do elemental ratio analysis? Please provide EDS analysis of the as-synthesized sample and give more explanation.
  3. Please provide more characteristic results of the synthesized samples to compare with other photocatalysts (such as Cds or MoS2). The author should also explain the characteristics of the ternary heterojunctions photocatalyst.
  4. In Figure S1(b), The labels of the X-axis is not showing up.

Round 2

Reviewer 2 Report

The revised manuscript is satisfactorily addressed the reviewer's comments and well improved. The present reviewer recommends this manuscript for publication in Nanomaterials.